# Characterization and whole genome sequencing of *Saccharomyces cerevisiae* strains lacking several amino acid transporters: Tools for studying amino acid transport

Unnati Sonawala¤a, Aymeric Busidan¤b, David Haak, Guillaume Pilot ![ORCID]*

School of Plant and Environmental Sciences, Virginia Tech, Blacksburg, Virginia, United States of America

¤a Current Address: Crop Science Centre, University of Cambridge, Cambridge, CB3 0LE UK
¤b Current Address: Department of Biology and Biological Engineering, California Institute of Technology, Pasadena, CA 91125, USA

* gpilot@vt.edu

## Abstract

*Saccharomyces cerevisiae* mutants have been used since the early 1980s as a tool for characterizing genes from other organisms by functional complementation. This approach has been extremely successful in cloning and studying transporters; for instance, plant amino acid, sugar, urea, ammonium, peptide, sodium, and potassium transporters were characterized using yeast mutants lacking these functions. Over the years, new strains lacking even more endogenous transporters have been developed, enabling the characterization of transport properties of heterologous proteins in a more precise way. Furthermore, these strains provide the added possibility of characterizing a transporter belonging to a family of proteins in isolation, and thus can be used to study the relative contribution of redundant transporters to the whole function. We focused on amino acid transport, starting with the yeast strain 22Δ8AA, which was developed to clone plant amino acid transporters in the early 2000s. We recently deleted two additional amino acid permeases, Gnp1 and Agp1, creating 22Δ10α. In the present work, five additional permeases (Bap3, Tat1, Tat2, Agp3, Bap2) were deleted from 22Δ10α genome, in a combination of up to three at a time. Unexpectedly, the amino acid transport properties of the new strains were not very different from the parent, suggesting that these amino acid permeases play a minor role in amino acid uptake, at least in our conditions. Furthermore, the inability to utilize certain amino acids as sole nitrogen source did not correlate with reduced uptake activity, questioning the well-accepted relationship between lack of growth and loss of transport properties. Finally, in order to verify the mutations and the integrity of 22Δ10α genome, we performed whole-genome sequencing of 22Δ10α using long-read PacBio sequencing technology. We successfully assembled 22Δ10α's genome *de novo*, identified all expected mutations and precisely characterized the nature of the deletions of the ten amino acid transporters. The sequencing data and genome

**Data availability statement:** The sequencing data and final genome assembly have been submitted under the NCBI BioProject, ID PRJNA862461. All other relevant data are within the manuscript and its Supporting Information.

**Funding:** National Science Foundation of USA - Grant IOS-1353366 to GP Hatch Program of the National Institute of Food and Agriculture of USA ) and the Virginia Agricultural Experiment Station - Grant VA-135908 for GP The funders had no role in study design, data collection and analysis, decision to publish, or preparation of the manuscript

**Competing interests:** The authors have declared that no competing interests exist.

will serve as a valuable resource to researchers interested in using these strains as a tool for amino acid transport study.

## Introduction

Fundamental cellular processes and metabolic pathways are well conserved among eukaryotic organisms, enabling researchers to use simpler eukaryotic organisms to study protein function in more complex organisms. Heterologous expression of a gene allows to study the biochemical function and properties of the encoded protein and further understand its function [1]. The function of a gene can be better characterized by expression in a mutant background where the genes responsible for the same activity have been inactivated. Such a strain exhibits a phenotypic defect, which is reverted by the expression of the gene of interest, a method called functional complementation of the yeast mutant [2,3]. Amongst the many hosts used for functional complementation, *Saccharomyces cerevisiae* stands out as an excellent model organism for expressing plant proteins and membrane transporters in general [2,4–6].

Amino acids are critical for many cellular processes such as nitrogen homeostasis, protein synthesis, and nucleoside synthesis. Amino acid transporters, which mediate the translocation of amino acids across membranes, are *bona fide* components of metabolic pathways [7] and thus play a fundamental role in these functions. Numerous membrane proteins, including amino acid transporters from both plants and animals, have been characterized through expression in yeast cells [2]. Studying amino acid transporters by functional complementation of yeast transport mutants is achieved by testing the growth of yeast on a medium containing amino acids as the sole nitrogen source: the yeast can take up nitrogen (provided by the amino acid, [8]) only when the amino acid transport function is provided at the plasma membrane by the foreign gene. Alternatively, amino acid uptake of the expressed transporter can directly be screened and measured by determining the amount of radiolabeled amino acids taken up by cells when provided in the external medium [9,10].

*Saccharomyces cerevisiae* contains several endogenous amino acid permeases, 22 of which are localized to the plasma membrane [8,11]. They all belong to the APC (Amino acid-Polyamine-organo Cation) superfamily and are further divided into the YAT (Yeast Amino acid Transport), LAT (L-type Amino acid Transport), and ACT (Amino acid Choline Transporter) families [12]. Some of these transporters display broad specificities for amino acids, whereas others are more specific and transport only a few amino acids [8,13,14].

Functional complementation of yeast was first used to identify the amino acid transporters from the plant *Arabidopsis thaliana* in the early 1990s. The yeast strain JT16 lacking both the histidine permease (Hip1) and an enzyme required in the synthesis of histidine (His4) was used to screen for a plant amino acid transporter that could take up histidine [15]. Around the same time, yeast strain 22574d (Fig 1)

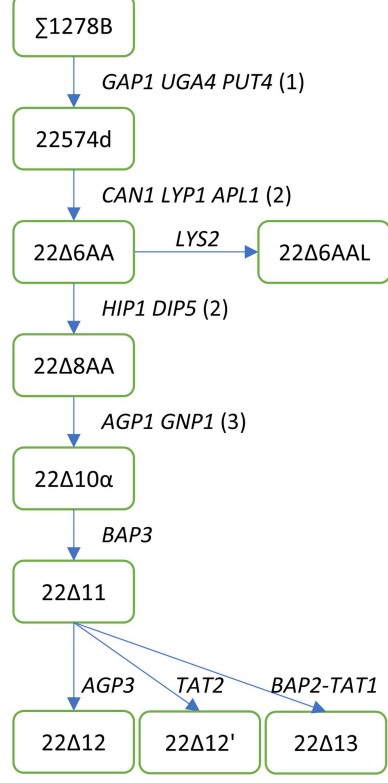

**Fig 1. Schematic of lineage leading to yeast strain 22Δ10α and its progenies.** The major strain names are indicated in green boxes and the knock-out of genes encoding given amino acid permeases are noted beneath the arrows. References: (1) [41]; (2) [23]; (3) [25].

lacking a broad specificity permease (Gap1), a γ-aminobutyric acid (GABA) permease (Uga4) and a high-affinity proline permease (Put4) was used to identify a plant amino acid transporter that was able to transport proline [16]. Both groups had simultaneously cloned the first secondary-active amino acid transporter, AtAAP1 (*Arabidopsis thaliana* Amino Acid Permease 1) [17]. Several other plant amino acid transporters were identified in the second half of the 90's by using complementation of these yeast mutants lacking their endogenous amino acid permeases [18]; see [10,19,20] for identification of other AtAAPs and related proline transporters, [21] for identification of cationic amino acid transporter (AtCAT1) and [22] for identification of lysine-histidine transporter (AtLHT1). The mutant yeast strain 22574d is still able to transport other amino acids apart from GABA and proline by using the remaining endogenous amino acid permeases. Fischer et al. (2002) [23] further deleted three other yeast amino acid permease genes responsible for arginine (*CAN1*) and lysine (*LYP1* and *ALP1*) transport. This mutant yeast strain was named 22Δ6AA (after the 6 deleted amino acid permeases). Deletion of the gene encoding the Lys biosynthesis gene *LYS2* in 22Δ6AA led to the strain 22Δ6AAL, enabling sensitive detection of Lys transport activity by complementation. To study the transport of other amino acids, two other permease genes were deleted in the 22Δ6AA background (*HIP1* and *DIP5* to disrupt histidine, and glutamate/aspartate transport respectively) leading to the strain 22Δ8AA [23] (Fig 1). 22Δ8AA is unable to grow on Asp, Arg, citrulline (Cit), GABA, Glu and Pro as sole nitrogen source supplied at 3 mmol.l$^{-1}$. This strain is still able to grow on 13 other amino acids. We previously reported deleting *AGP1* and *GNP1* (whose encoded proteins are necessary for the uptake of Threonine [24]) in the 22Δ8AA background to increase the number of amino acids whose transport can be studied, creating 22Δ10α [25].

22Δ10α was found to be unable to grow on 10 additional amino acids as its sole nitrogen source compared to 22Δ8AA [25], probably because Agp1 is actually a broad specificity amino acid permease [26].

In the present work, we wanted to reduce the background transport for some aromatic amino acids by deleting five other endogenous amino acid permeases in the 22Δ10α background in three different combinations. We also tested if the resulting strains could be used to study transport at concentrations higher and lower than 3 mmol.l⁻¹, which would be useful in characterizing low- or high-affinity plant amino acid permeases. Finally, the genome of 22Δ10α was sequenced and analyzed to identify any major chromosomal rearrangement that could have arisen during the successive gene deletion events.

## Materials and methods

### Yeast strains and manipulation

Our findings demonstrate that the mating type of 22Δ10α and its progenitor 22Δ8AA is MAT*a*; the genotype of both strains has thus been corrected in this Material and Method section. See Table 1 for the genotypes of the strains used in this work. Note that 23344c is not a progenitor of 22Δ8AA but it has been typically used as a positive control in complementation experiment because it does not lack any of the amino acid permeases. Like the 22Δ series of strains, 23344c is a descendant of ∑1278b. Yeast cells were transformed using the lithium acetate method [27], and genes were deleted from the genome sequentially using the insertion of the kanMX cassette flanked by two loxP sites [28].

### Cloning

ScGap1 was PCR-amplified from *Saccharomyces cerevisiae* genomic DNA, cloned into pDONRZeo using the Gateway technology (ThermoFisher Scientific), and moved to the destination vector pRS-Ura-Ws (a derivative of pRS416, in which the gateway-compatible expression cassette from pDR196-Ws was inserted).

### DNA extraction, sequencing and genome sequence analysis

Yeast cells were grown in 50 ml of YPDA medium (10 g/l yeast extract, 20 g/l bacto peptone, 20 g/l glucose, 80 mg/l adenine) (starting at OD of 0.1 at 600 nm) until they reached the exponential phase (two doublings), washed and resuspended in 3 ml of 1 M sorbitol. Cells were then converted to spheroplasts through treatment with 50 μl of a solution of a 4 mg/ml Zymolyase 100T (United States Biological) in 50 mM Tris-HCl, pH 7.5, 1 mM EDTA and 50% glycerol, and incubating at 30°C with gentle shaking (70 rpm) for 1 hour, followed by a wash with 1 M Sorbitol. High molecular weight DNA

**Table 1. Yeast strains produced and used in this work.**

| Strain | Genotype | Reference |
|--------|----------|-----------|
| 23344c | *MATα ura3–52* | [29] |
| 22Δ8AA | *MATa gap1–1 put4–1 uga4–1 can1::HisG lyp1-alp1::HisG hip1::HisG dip5::HisG ura3–1* | [23] |
| 22Δ10α | *MATa gap1–1 put4–1 uga4–1 can1::HisG lyp1-alp1::HisG hip1::HisG dip5::HisG gnp1Δ agp1Δ ura3–1* | [25] |
| 22Δ11 | 22Δ10α *bap3Δ* | This work |
| 22Δ12 | 22Δ10α *bap3Δ* agp3*::kanMX* | This work |
| 22Δ12' | 22Δ10α *bap3Δ tat2::kanMX* | This work |
| 22Δ13 | 22Δ10α *bap3Δ bap2-tat1::kanMX* | This work |
| AH109 | *MATa trp1–901 leu2–3, 112 ura3–52 his3–200 gal4Δ gal80Δ LYS2::GAL1$_{UAS}$-GAL1$_{TATA}$-HIS3 GAL2$_{UAS}$-GAL2$_{TATA}$-ADE2 URA3::MEL1$_{UAS}$-MEL1$_{TATA}$-lacZ* | Clontech |
| Y187 | *MATα ura3–52 his3–200 ade2–101 trp1–901 leu2–3, 112 gal4Δ met⁻ gal80Δ URA3::GAL1$_{UAS}$-GAL1$_{TATA}$-lacZ* | [30] |

was extracted from the pelleted spheroplasts using MagAttract HMW DNA kit (Qiagen) by following the manufacturer's instructions. DNA was sequenced by Novogene on the PacBio platform. Continuous long reads were assembled *de novo* using canu (version 1.9) with the additional parameters of *minReadLength = 5000* and *minOverlapLength = 1000* [31]. Reference-guided contig rearrangement and scaffolding was performed using RagTag [32]. Assembly statistics were generated using QUAST [33]. The genome sequence for the yeast strain S288C was obtained from yeastgenome. org (version R-64-1-1) and used as a reference genome. Reads were mapped to S288C via minimap2 with the option *-ax map-pb*. NGMLR mapped reads and Sniffles (1.0.12) were used for calling structural variants [34]. Variant calls were restricted to reads with mapping quality >30 using the parameter *-q 30*. To get high confidence structural variants the output from Sniffles was filtered for variants with read support of greater than 30 reads and alternate allele frequency greater than 0.4. Alignments and variants were visualized with Integrative Genomics Viewer (version 2.7) [35].

## Yeast growth and radioactive uptake assays

Amino acid uptake assays were performed as described by [36] using $^3$H-labelled amino acids. Briefly, yeast cells were grown overnight in SD medium (6.7 g/l Yeast Nitrogen Base without amino acids (Difco), 20 g/l glucose, supplemented with amino acids but lacking uracil, pH 6.3) overnight at 30˚C. They were then subcultured in 15 ml of SD medium at an OD of 0.1 and incubated at 30˚C until they reached an OD of ~0.5. Yeast cells were then pelleted at 2,500g and washed by resuspending in water and centrifuging at 2,500g. Washed cells were resuspended in uptake buffer (50 mM $KH_2PO_4$ and 600 mM sorbitol at pH 4.5) at an OD of 5. 50 µL of these cells were aliquoted to be used for uptake per replicate of each amino acid and placed on ice until use. For the assay itself, 5 µL of 1 M glucose was added to 50 µL of the aliquoted cells in uptake buffer and incubated at 30˚C in a thermal mixer for 5 minutes. Exactly 5 minutes later at 30˚C, 55 µL of a mix of the unlabeled amino acid at the required concentration and 1 µCi of $^3$H-labelled amino acid in uptake buffer was added, and the resulting mix placed back on the thermal mixer for exactly 3 minutes. The cells were filtered using 24 mm Whatman filters (cat no. 1822–024) using a filtration manifold (DHI lab Filtration Manifold 10 x 20ml, cat no. EQU-FM-10X20-SET): the cells were transferred to 5 ml of uptake buffer placed in the filtering device; the cells and buffer were drained through the filter under vacuum; 5 ml of uptake buffer was added and drained similarly. The filters were then added transferred to scintillation vials, filled with 4 ml of Ultima Gold XR (Revvity).

## Complementation assays

Yeast strains were grown overnight in SD medium (1.7 g/l Yeast nitrogen base with ammonium sulfate, 20 g/l glucose pH 6.3). Yeast cells were diluted to the appropriate OD and 4 µL drops were laid on a minimum medium [37] supplemented with the specified concentration of the mentioned amino acids at 30˚C.

## Yeast doubling time measurements

For measuring the growth rate and doubling of yeast strains, three independent 5 ml cultures in YPDA were grown overnight at 30˚C and used the next morning to start a subculture in either YPDA or SD (+uracil) medium at OD of 0.1. The subculture was allowed to grow until an OD of about 0.5 and used to start 200 µL cultures in 96-well plates at an OD of about 0.05-0.1 in either YPDA or SD (+uracil), as specified. Each biological replicate was also technically replicated twice on the 96-well plate. A synergy HTX plate reader (Biotek) was used for measuring the OD of the cultures with the following settings: temperature set at 30˚C, measurements taken every 5 min at 600 nm, continuous orbital shaking at 559 nm (slow speed) for 15 hours. The growth curve from the data was fit to logistic equation and growth characteristics were measured using the Growthcurver package in R [38].

## Determination of mating type

Genomic DNA extracted from yeast grown overnight in YPDA was analyzed with specific oligonucleotides as described in [39]. For the mating test, 23344c, 22Δ8AA, 22Δ10α, AH109 and Y187 were grown on solid YPDA and resuspended

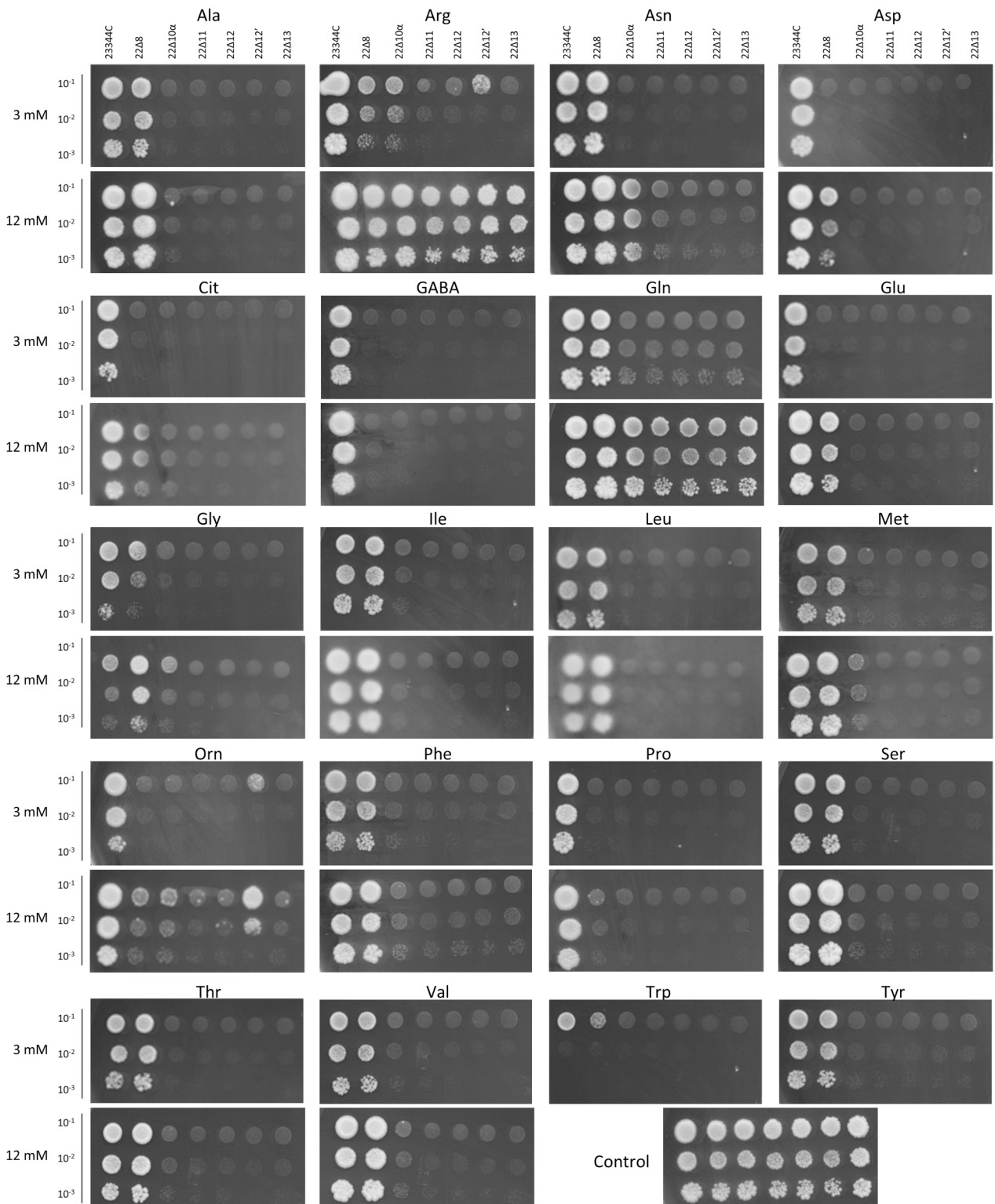

**Fig 2. Growth comparison of 23344C, 22Δ10α, 22Δ11, 22Δ12, 22Δ12' and 22Δ13 on amino acids as sole nitrogen source.** OD for each strain was adjusted to 0.1, 0.01 and 0.001. Drops of 4 μL were aligned on minimum medium containing the labeled amino acid at 3 or 12 mmol.l⁻¹ as sole nitrogen source. The control medium contained 1.5 mM $(NH_4)_2SO_4$. Growth was not tested on 12 mM Trp or Tyr because these amino acids are not stable at this concentration. Pictures were taken after 4 days at 30°C.

in 200 μl water. About 50 μl of each solution was mixed as described in the figure legend, and 10 μl were dropped on a YPDA plate. After overnight growth at 30°C, cells were resuspended in 100 μl water, and streaked on SD media supplemented with the compounds indicated in the figure legend, and grown for three days at 30°C.

## Results

### Deletion of amino acid permeases in 22Δ10α

Previous work by Regenberg et al. [40] and Bianchi et al. [8] showed that the endogenous permeases Bap2, Bap3, Tat1 and Tat2 are important for the transport of aromatic amino acids (Phe, Trp and Tyr), for transport of the branched-chain amino acids (Leu, Ile and Val) and to a lesser degree Ala and Gly across the plasma membrane. Agp3 has been shown to transport leucine and becomes more important for yeast growth in low nutrient conditions or in yeast mutants lacking the broad-specificity permeases Gap1 and Agp1 [41]. Based on their importance in amino acid uptake by yeast cells, these five endogenous amino acid permeases were deleted from the genome of 22Δ10α in the following combinations: deletion of *BAP3* leading to 22Δ11, deletion of *BAP2* and *TAT1* (these genes are next to one-another on the chromosome) in the 22Δ11 background leading to 22Δ13, and deletion of *AGP3* or *TAT2* in the 22Δ11 background each leading to 22Δ12 or 22Δ12' respectively (Fig 1). A few attempts were made to make further deletions in the 22Δ13 background but they were unsuccessful.

### Characterization of the transport ability of the 22Δ11, 22Δ12, 22Δ12' and 22Δ13 strains

To test if the additional gene deletions would reduce the background growth of the resulting strains on amino acids as sole nitrogen source, 23344c, 22Δ8AA, 22Δ10α, 22Δ11, 22Δ12, 22Δ12' and 22Δ13 cells were grown on minimum medium containing uracil with each amino acid supplied at 3 mmol.l$^{-1}$ as the sole nitrogen source. Compared to 23344c, 22Δ8AA was, as expected, unable to use Asp, Cit, GABA, Glu, Gly, Ornithine (Orn) and Pro as a nitrogen source (Fig 2) [23]. In addition to those amino acids, 22Δ10α was unable to use Ala, Asn, Gln, Gly, Ile, Leu, Met, Phe, Ser, Thr, Val Trp and Tyr as a nitrogen source (Fig 2; S1 Fig) [25]. Compared to 22Δ10α, further gene deletions had little effect on the background growth of the cells, except for Arg when supplied at 3 mmol.l$^{-1}$, and Met supplied at 12 mmol.l$^{-1}$(Fig 2 and S1 Fig). The ability of the growth defect to be functionally complemented by amino acid transporters expressed from a plasmid was tested by expressing the ScGap1 amino acid permease [42] in 22Δ10α, 22Δ11, 22Δ12' and 22Δ13. The growth was compared to cells transformed with an empty plasmid. For all tested amino acids and concentrations (0.5, 3, 9 or 12 mM), ScGap1 enabled a similar growth for each of the three strains, well above the background. In this growth experiment, the background growth on Met was reduced in 22Δ13 compared to the other three strains (S2 Fig).

The sensitivity of the growth complementation assay is limited by the concentration of the supplied amino acid. Activity of low-capacity transporters or transporters not well expressed at the plasma membrane would indeed not lead to any noticeable growth when the concentration of the supplied amino acid is less than 0.5 mM. To test whether such a protein displays amino acid transport activity, a more suitable approach consists of directly measuring the uptake of radiolabeled amino acid, over the course of a few minutes. In this assay, even minute uptake can be measured because of the sensitivity of the instruments to detect radioactivity. We tested whether deletion of amino acid permeases in 22Δ10α lowered the background amino acid uptake when amino acids were supplied at very low concentrations (3, 30 or 300 μM). The uptake of Asp, Gln, Glu, Lys, Met and Val supplied at any of the three concentrations was lower for 22Δ10α and 22Δ13 than for 23344c (Fig 3). Surprisingly, the uptake of Leu, Pro and Trp was not reduced in 22Δ10α and 22Δ13 compared to 23344c. Overall, the background amino acid uptake was similar between 22Δ10α and 22Δ13, despite the three additional permease genes deleted (Fig 3).

### Sequencing and analysis of 22Δ10α genome

The mutant yeast strains originating from 22574d have been used to study plant amino acid transporters for several years. However, apart from the parent Σ1278b, none of these have been subjected to whole genome sequencing. Previous

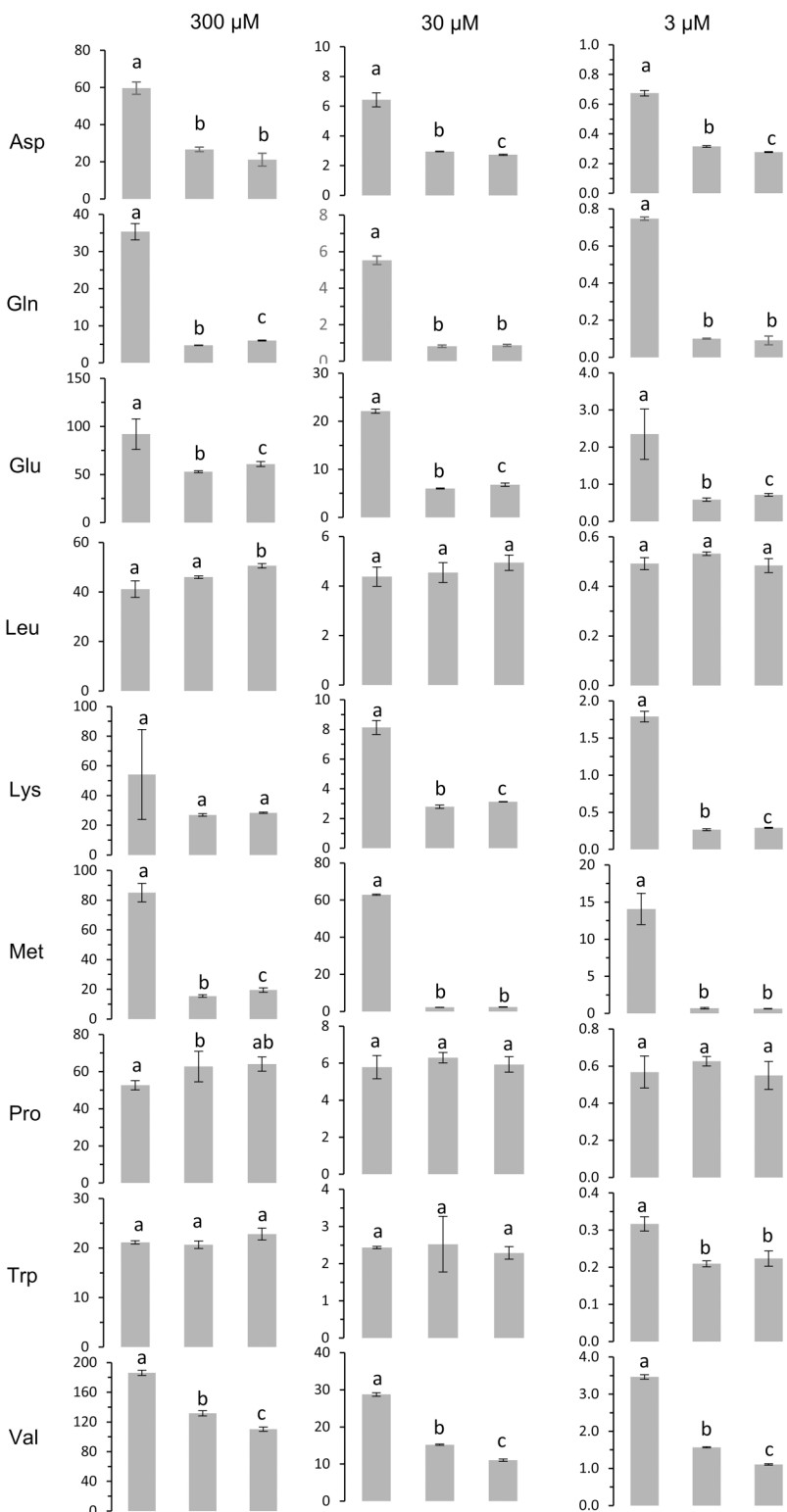

**Fig 3. Uptake of radiolabeled amino acids by 23344C, 22Δ10α and 22Δ13 cells.** Uptake of amino acids supplied at concentrations of 300 μM, 30 μM and 3 μM by control yeast (23344c) and amino acid permease mutants (22Δ10α and 22Δ13) was measured after 3 minutes. Error bars represent standard deviation among technical replicates (n = 3). Different letters indicate statistical significance at p ≤ 0.05 according to t-test with Holm-Bonferroni correction for multiple comparisons.

(a)

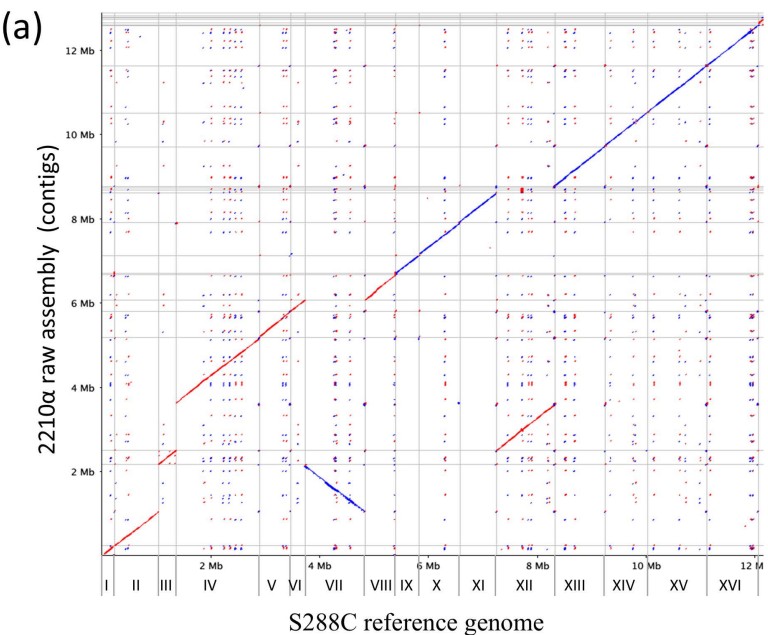

(b)

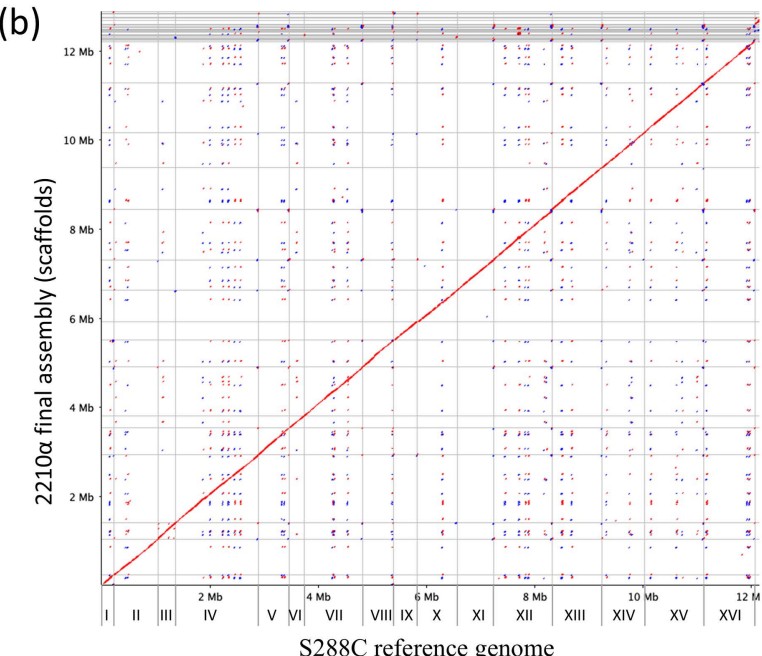

**Fig 4. Whole genome alignment of 22Δ10α to S288C genome.** Whole genome alignment of raw (a) and of the final (b) genome assembly of 22Δ10α against the S288C reference genome. Chromosomes are designated with Roman numerals.

**Table 2. Summary of the observed indels in the 11 genes inactivated in 22Δ10α.**

| Genes | 5' Location[†] | Deletion length[‡] | | Insertion length | Inserted element[*] |
|---|---|---|---|---|---|
| | | Observed (if any) | Expected (if known) | Observed[§] | |
| URA3 | 369 | 1 | | | |
| GAP1 | −1103 | ~4000 | | ~9000 | Transposon |
| PUT4 | 5 | | | ~6000 | Transposon |
| UGA4 | 1182 | | | ~6000 | Transposon |
| CAN1 | 681 | | | ~250 | hisG scar |
| ALP1-LYP1 | 835 | ~2500 | | ~1200 | hisG scar |
| HIP1 | 1254 | ~250 | | ~1000 | hisG scar |
| DIP5 | 538 | | | ~1200 | hisG scar |
| AGP1 | −63 | ~2000 | 2.012 | 34 | LoxP scar |
| GNP1 | −61 | ~2000 | 2.128 | 34 | LoxP scar |

[†]Approximate distance in bp from the ATG (ATG of *ALP1* for the *ALP1-LYP1* locus)

[‡]Approximate size of the deletion in kbp, using the S288C genome sequence as a reference. No value if no deletion was observed or if the expected size of the deletion was unknown.

[§]Approximate size of the inserted DNA sequence in bp

[*]"Transposon", transposon-related sequence; "hisG scar", leftover sequence from the deletion approach [23]; "loxP scar" leftover loxP site from the deletion approach [25].

deletion studies, such as those for yeast deletion collection, have found that deleting yeast genes can cause off-target deletions or gene duplications, and sometimes chromosomal rearrangements [43,44]. Although rare, such collateral changes and off-target effects to the genome are possible. In order to aid future research with these mutant strains, we sequenced the 22Δ10α genome rather than its descendants since they did not show much improvement in background growth. Using 8.12 Gbp of long-read PacBio sequencing data, the 22Δ10α genome was assembled *de novo* using Canu to an estimated depth of 600x [31], leading to 25 contigs. Aligning the contigs to the S288C reference genome showed coverage over all chromosomes further supporting the assembly (Fig 4a). Furthermore, after reference-guided separation of erroneously collapsed contigs, scaffolding and polishing, no large duplications or translocations were identified (Fig 4b), leading to a 12.87 Mbp assembly in 48 scaffolds. Noticeably, 95% of the assembly was contained within the largest 16 scaffolds (12.2 Mbp, corresponding to the entire yeast genome), with the remaining 5% corresponding to totally ordered contigs not assembled into scaffolds, and aligning with miscellaneous regions of the genome (Fig 4b and S4 Fig).

Structural variants (SV) were identified from 22Δ10α reads aligned to the S288C reference genome and used to identify structural variants (SV) using an SV caller specifically designed for long-read sequences with higher error rates. S288C genome was used for this purpose given its highly-contiguous chromosome-level assembly compared to the more fragmented Σ1278b assembly [45,46]. SV calls were filtered to get high-confidence variants (see methods), resulting in 221 nuclear SVs (S1 Data). The majority of the SVs were associated with transposable elements (TEs) and long-terminal repeats (LTRs) (S1 Data). We did not observe any insertion related to the hisG or KanMX/loxP deletion cassettes outside of the expected 7 amino acid permease genes (see below).

We then identified the mutations of the 11 inactivated genes (*URA3* and 10 amino acid permeases), corresponding to loxP deletions, hisG insertions or other (the nature of the mutations of *URA3*, *GAP1*, *PUT4* and *UGA4* have not been reported). In the case of insertions, reads spanning the expected gene were used to identify the insert using BLAST. Indels were found in all the ten amino acid permease genes, therefore confirming and identifying the nature of these mutations (Table 2, S3 Fig). The indel in *GAP1* resulted from a ~4 kbp deletion replaced by a ~6 kbp insertion of a Ty1 LTR retrotransposon; ~6 kbp insertions of Ty1 elements were also found at the start of *PUT4* and within *UGA4* [42].

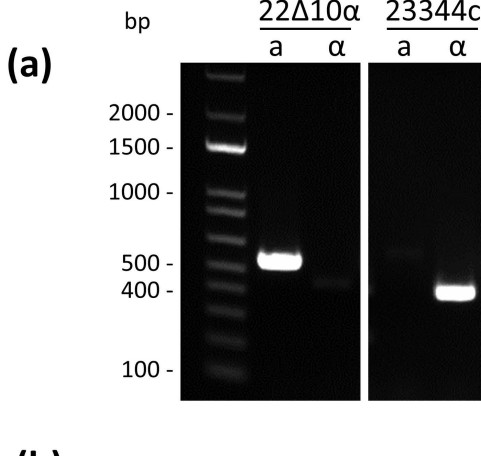

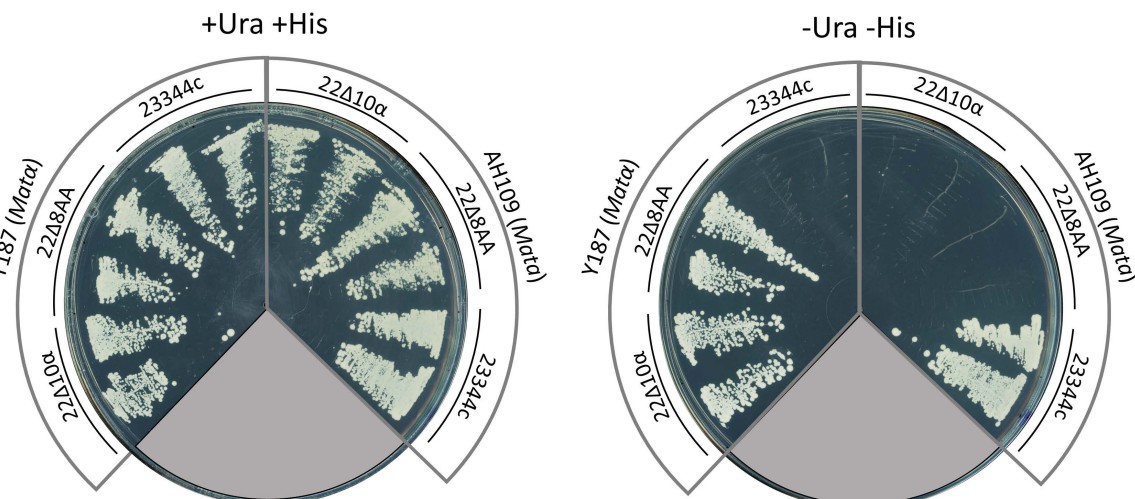

**Fig 5. Mating type genotype and assay of 22Δ10α.** (a) Genotyping PCR of 22Δ10α and 23344c strains with oligonucleotide pairs that correspond to the Mat*a* (a) or Mat*α* (α) mating type. Expected sizes are Mat*a*: 544 bp and Mat*α*: 404 bp. (b) Mating assay between 23344c, 22Δ8AA and 22Δ10α with Y187 and AH109. Mated cells were streaked on SD medium supplemented with Met, Ade, Trp, Leu, and with or without Ura and His. Diploid cells, resulting from mating of cells of different mating types should be both Ura and His prototroph. Two colonies of each of the tested strains were mated with AH109 and Y187.

*CAN1*, *HIP1*, *ALP1-LYP1* and *DIP5*, which were all deleted using the *hisG-URA3-neo-hisG* cassette [23], were found to have insertions ranging from ~250 bp to 2.5 kbp left by the hisG cassette replacing the genes. *AGP1* and *GNP1*, which were deleted in our laboratory using the *loxP-kanMX-loxP* cassette [25], were both confirmed to have the expected 2 kb-deletions abrogating the entire locus for each of these genes. Finally, analyzing the sequence of the *URA3* CDS, we found a 1 bp-deletion at position 116,535 of chromosome V (Table 2, S3j Fig), which is likely the reason of the *ura3–1* mutation leading to uracil auxotrophy.

Genome sequencing also revealed that the mating locus of 22Δ10α was Mat*a*, while this strain has always been assigned with MATα in the literature (see [23]). To experimentally determine 22Δ10α mating type, the Mat locus was genotyped by PCR, confirming the sequencing data, being indeed Mat*a*. We confirmed the genotype of 23344c, also a

descendant of ∑1278b (MAT*a*), as MAT*α* (Fig 5a). In addition, 22Δ8AA and 22Δ10α mated with Y187 (MAT*a*) but not with AH109 (MAT*a*), while the reverse was observed for 23344c (Fig 5b), proving that the phenotypes of 22Δ8AA and 22Δ10α are Mat*a* and that of 23344c is MAT*α*. The mating type of 22Δ10α is therefore Mat*a* (note that the "α" of 22Δ10α is not related to its mating type, but to its order in our deletion procedure).

## Analysis of growth speed of 22Δ10α, 22Δ11 and 22Δ13

It was noticed that 22Δ11 and 22Δ13 grew appreciably slower compared to 22Δ10α and sometimes led to poor trans-formation efficiency. The doubling time was therefore calculated in both nutrient-rich YPDA medium and the minimal SD medium (S5 Fig). In YPDA 22Δ11, 22Δ12, 22Δ12' and 22Δ13 doubling time was 25–30 min longer than the control strain 23344c. The impeded growth was magnified when these mutants were grown in minimal SD medium with these strains doubling up to 40–50 minutes slower than 23344c. Later analysis of sibling colonies kept frozen during the generation of 22Δ11 showed that the slow growth phenotype did not happen during the insertion of the kanMX cassette, but in the step following the cre-lox recombination used to remove the cassette. This step corresponded to the curing of the pSH47 plas-mid, and about half of the tested colonies displayed the same long doubling time as 22Δ11, while the other colonies grew similar to 22Δ10α. In addition, genotyping the deletion sites of *GNP1*, *AGP1* and *BAP3* in 22Δ11 siblings or descendants did not provide evidence of any genome rearrangement that could have arisen by recombination between distant loxP sites (data not shown).

## Discussion

### Creation of a suite of yeast strains deleted for up to 13 amino acid permeases

Here we report the characterization of yeast strains deleted for the main plasma membrane amino acid permease genes, focusing on their ability to grow on amino acid as the sole nitrogen source. These strains have been developed from the well characterized [25] and extensively used [47–56] 22Δ10α strain by deleting additional amino acid permease genes. We generated 22Δ11, 22Δ12, 22Δ12' and 22Δ13 to reduce the background growth of the 22Δ10α on selective amino acid media. Unfortunately, we did not observe significant reduction of background: all strains grew similarly on most of the tested media (Fig 2). The only improvement was observed when deleting BAP3 from 22Δ10α to create 22Δ11. This strain and its descendants showed deficiency in growth on 3 mM Arg as the sole nitrogen source when compared to 22Δ10α (Fig 2). The modest reduction in background growth observed in 22Δ13 on 9 and 12 mM Met compared to the other strains suggests that Bap2 and Tat1 facilitate Met uptake when provided at more than 3 mM outside of the cell (S2 Fig).

We initially followed up with deleting additional genes and intended to delete *TAT2* from 22Δ13 as an avenue to decrease the background growth further, but with no success. Because the kanMX cassette had not been removed from the 22Δ13 genome, we used the natNT2 cassette which provides resistance to the antibiotic cloNAT [57], thereby skip-ping one step in the process of creating higher order mutants. While the *TAT2* deletion leads to viable strains, proven by the viability of 22Δ12', it is possible that our conditions were not optimal for selecting the cells which successfully inserted the cassette into *TAT2*, or that the double mutant *TAT1 TAT2* in the 22Δ11background is not viable. We did not pursue this further because the higher order deletion mutants 22Δ11, 22Δ12, 22Δ12' and 22Δ13 did not show noticeable growth differences compared to 22Δ10α on selective amino acid media (Fig 2).

The main limitation of the use of these higher order deletion mutants is their higher doubling time compared to 22Δ10α. Our tests of 22Δ11, its sibling, and descendants did not provide any information about the reason for this effect, except that it happened during the curing of pSH47, the plasmid that encodes the CRE recombinase. The longer doubling time is thus not caused by deletion of *BAP3*. While the slower growth makes working with the strains less convenient, this does not, a priori, affect their potency as a tool for characterizing amino acid transport and transporters. At least, this work shows that deletion of up to 13 amino acid permeases is viable. Researchers wanting to re-create the higher order dele-tion mutants could start from one of the 22Δ11 siblings that does not show this defect (available upon request).

For the sole purpose of functional complementation, 22Δ10α is therefore the strain of choice: it has the least number of deleted amino acid permeases; it grows at a similar speed as the parent strains 23344c and 22Δ8A; its genome is sequenced (this study, see below); and, most importantly, its phenotype on amino acid selective media is similar to the higher order deletions mutants. These latter mutants could nevertheless be useful for yeast researchers studying the function of yeast amino acid permeases or their regulation, because they correspond to mutants of *AGP3*, *TAT2* and *BAP2-TAT1* in the same 22Δ11 background: detailed characterization of the strains could reveal specific roles for those genes. Since four out of the 19 plasma membrane permeases mediating amino acid transport have not been deleted in our suite of strains, namely *AGP2, MUP1, MUP3, YCT1 and VBA5,* future work could use our strain as starting material for their characterization.

### Genome sequencing confirms the nature of the deleted genes in 22Δ10α and corrects the mating type of the strains

For the reasons provided above, 22Δ10α is the strain that will be the most used for complementation assays. We thus decided to sequence its genome to define the nature of the gene inactivations and test whether any major structural rearrangement had occurred during the successive gene deletions. Several structural variants were observed and corresponded to transposons. We compared 22Δ10α genome sequence to the S288C reference genome, because of its higher quality compared to that of the parental strain Σ1278b. Therefore, many of these structural variants could stem from the difference between Σ1278b and S288C. We did not find any off-target deletions/scars arising from the successive manipulations of the genome, confirming that the phenotype of 22Δ10α is a consequence of the deletion of the 10 amino acid permeases and no other gene.

Genome sequencing identified the nature of the *ura3-1* mutation and corrected the mating type of 22Δ10α. The lineage of 22Δ10α originated from yeast strains published in 1970 and 1987 that were mated to create 22574d [42], the direct progenitor of 22Δ6AA, 22Δ8AA [23] and finally 22Δ10α [25]. 22574d carries the same *ura3-1* mutation as MG471 [42], which was likely created by ethyl methane sulfonate (EMS) mutagenesis [58]. We identified a deletion at position 369 in the *URA3* gene leading to a frame shift. While EMS is typically known for creating G/C-to-A/T transitions, there is evidence that it can lead to both small and large indels or even chromosome breaks, albeit at a low frequency [59–61]. We conclude that the *ura3–1* mutation in this family of strains corresponds to a deletion of a thymidine at position 369 in *URA3*. Analysis of the genome sequence and further studies (Fig 5) revealed that 22Δ10α's mating type is Mat*a*, contrary to what can be found in the literature, where 22574d and its descendant are marked as being Mat*α* [23,42]. The mating type of 22Δ10α's parental strain, 22Δ8AA, is also Mat*a*. It is unknown at which step of the creation of 22Δ8AA the change from Matα to Mat*a* occurred, and if any of the other strains were tested for mating type at any time. This information is relevant for researchers wishing to mate this strain to combine mutations from other strains.

### Discrepancy between the growth phenotype on amino acid selective media and radioactive amino acid uptake results

Finding a *Saccharomyces cerevisiae* strain lacking an amino acid permease gene and unable to grow on an amino acid as the sole nitrogen source should logically display decreased amino acid uptake ability. This principle is fundamental to the functional complementation that aims to isolate genes from other species endowed with the same function. The reduced growth of 22Δ10α and 22Δ13 on Asp, Gln, Glu, Lys, Met and Val selective media was paralleled by a decreased uptake measured using radiolabeled amino acids in 3 min (Figs 2, 3 and S2 Fig). This relationship was not observed for Pro, Leu and Trp: the uptake was similar between 23344c, 22Δ10α and 22Δ13 (Fig 3), even though 22Δ10α and 22Δ13 were unable to grow on these amino acids (Fig 2). A critical methodological difference that needs consideration is that the concentrations used for the complementation assays in Fig 2 and the uptake experiments in Fig 3 differ by at least an order of magnitude (> 3mM and < 300 μM, respectively), which might explain the discrepancy. Nevertheless, 22Δ10α and

22Δ13 did not grow on Leu and Pro supplied at 0.5 mM (S2 Fig) while the corresponding uptakes at 0.3 mM were identical (Fig 3), suggesting that the reason appears to stem from biological rather than technical factors (see below). Because we have not measured the uptake of Ala, Arg, Asn, Cit, GABA, Gly, Ile, Lys, Orn, Phe, Ser, Thr and Tyr, similar discrepancies may exist for some of those amino acids.

Why cells that take up amino acid at the same rate as the wild type but cannot grow on this amino acid as the sole nitrogen source is difficult to explain. The simplest hypothesis is that the metabolic and transcriptomic states of the cells in the two assays are dramatically different, preventing the yeast from growing on amino acid selective medium even if the uptake of this amino acid is unaltered. Indeed, in the complementation assay, cells were grown on a solid medium that contains nitrogen in a slowly metabolizable form: amino acids need to be catabolized to release their nitrogen, most often ammonium, which is then assimilated into Gln, further used during the synthesis of the other amino acids. When cells are grown in liquid medium containing ammonium as the nitrogen source, like for uptake analysis, no amino acid needs to be degraded to release its nitrogen, which would conceivably lead to different activities of the various metabolic pathways. It has been noticed that the preferred source of nitrogen for ∑1278b is ammonium, contrary to S288C [62], suggesting different regulatory pathways for nitrogen utilization between ammonium and amino acids are at play. Apart from being metabolites, presence and concentration of amino acids in and outside the cells is sensed by numerous signaling cascades that control the expression of amino acid permeases and metabolic pathways [63]; Gap1, deleted in the 22Δ strain series, is one of such sensors [64]. The deletion of multiple permeases and sensors, the difference in nitrogen source and whether the medium is solid or liquid may therefore affect the metabolic pathways and the expression of the remaining amino acid transporters, such that the net uptake activity and nitrogen utilization are different in the two assay conditions. This phenomenon requires further investigation because it could reveal unsuspected mechanisms of regulation of amino acid permeases and metabolism. Metabolomics and proteomics, as well as measurement of amino acid uptake in yeast placed in presence of amino acids as sole nitrogen source for several hours might help unravel this discrepancy.

## Supporting information

**S1 Fig. Growth assay comparing growth of 23344C, 22Δ10α, 22Δ11, 22Δ12, 22Δ12' and 22Δ13 cells on given amino acid as sole nitrogen source.** Yeast cells were grown overnight in synthetic defined (SD) medium supplemented with uracil. OD for each strain was adjusted to 0.1, 0.01 and 0.001. Drops of 4 μL were aligned on minimum medium containing labeled amino acid at 3 (a) or 12 mmol/l$^{-1}$ (b) as sole nitrogen source. Pictures were taken after 2.5 days growth at 30˚C.
(PDF)

**S2 Fig. Functional complementation assay of 22Δ10α, 22Δ11, 22Δ12', and 22Δ13 yeast strains.** 22Δ10α, 22Δ11, 22Δ12', and 22Δ13 yeast strains were transformed with empty vector pRS-Ws or pRS-Ws vector containing the cDNA of GAP1 (General Amino acid Permease I, YKR039W). Yeast cells were grown overnight in selective medium. OD for each culture was adjusted to 1, 0.1 and 0.01. Drops of 5 μL were aligned on minimum medium containing amino acids at 0.5, 3, 9 or 12 mmol/l as sole nitrogen source, and grown at 30˚C. Picture shown were taken after the number of days indicated in the table. For each amino acid/ concentration combination, the dilutions of cells were dropped on a single Petri dish, despite them being shown as strips.
(PDF)

**S3 Fig. Integrated Genome Viewer images of indels in 22Δ10α.** (a-i) 22Δ10α PacBio sequencing reads were aligned to the S288C reference genome and were loaded in Integrative Genomics Viewer [35] along with the reference genome and its annotation. In mapped reads, deletions are indicated by a black line and insertions by purple regions (numbers

indicated length of deletion or insertion). When a read is clipped by more than 100 bp, the end of that read is marked in red. Indels of less than 30 bp are not labeled.
(PDF)

**S4 Fig. Assembly statistics of 22Δ10α genome.** 22Δ10α assembly statistics (left); plot of the cumulative length of the assembly vs. number of scaffolds (right).
(PDF)

**S5 Fig. Doubling time of yeast strains in YPDA and SD medium.** Yeast strains were grown in YPDA (a) or SD supplemented with URA (b) medium, and their ODs measured using a plate reader, every 5 min for 15 hours. The data were fitted to a standard form of logistic equations to get the growth characteristics including doubling time. Each boxplot the distribution of represents six data points per strain. * p-value<0.05.
(PDF)

**S1 Data. Text file of the sequence variants obtained from genome comparison of 22Δ10α and S288C.**
(ZIP)

**S1 Raw Image. Annotated, raw picture of the DNA gel for Fig 5a.**
(TIF)

## Acknowledgments

The authors thank Nima Trivedi for help in gene deletion and initial testing of the yeast strains on minimum media.

## Author contributions

**Conceptualization:** Unnati Sonawala, Guillaume Pilot.

**Investigation:** Unnati Sonawala, Aymeric Busidan.

**Supervision:** David Haak, Guillaume Pilot.

**Writing – original draft:** Unnati Sonawala.

**Writing – review & editing:** Unnati Sonawala, Guillaume Pilot.

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
