## [Decision Letter · Decision Letter 0]

6 Jan 2025

PONE-D-24-55165Characterization and whole genome sequencing of Saccharomyces cerevisiae strains lacking several amino acid transporters: tools for studying amino acid transportPLOS ONE

Dear Dr. Pilot,

Thank you for submitting your manuscript to PLOS ONE. After careful consideration, we feel that it has merit but does not fully meet PLOS ONE’s publication criteria as it currently stands. Therefore, we invite you to submit a revised version of the manuscript that addresses the points raised during the review process.

As you can see, the reviewers have made some suggestions to improve the manuscript. While I believe that the suggestion of running a mass spectrometry analysis has some value, I think It would not be required for publication in PLoS One. However, I do think that the suggestion about having a separate discussion section highlighting the major findings and limitations as well as future direction will be valuable for the readers who wish to use the strains in the future. While I opted for "major revisions" these changes should be easily implemented.

We look forward to receiving your revised manuscript.

Kind regards,

Patrick Lajoie, PhD

Academic Editor

PLOS ONE

Journal Requirements:

“National Science Foundation of USA - Grant IOS-1353366 to GP

Hatch Program of the National Institute of Food and Agriculture of USA ) and the Virginia Agricultural Experiment Station - Grant VA-135908 for GP”

“The authors thank Nima Trivedi for help in gene deletion and initial testing of the yeast strains on minimum media. This work was supported by the National Science Foundation of USA (Grant IOS-1353366 to GP) and the Hatch Program of the National Institute of Food and Agriculture of USA (VA-135908 for GP) and the Virginia Agricultural Experiment Station.”

“National Science Foundation of USA - Grant IOS-1353366 to GP

Hatch Program of the National Institute of Food and Agriculture of USA ) and the Virginia Agricultural Experiment Station - Grant VA-135908 for GP”

5. We note that your Data Availability Statement is currently as follows: [The sequencing data and final genome assembly have been submitted under the NCBI BioProject, ID PRJNA862461. All other relevant data are within the manuscript and its Supporting Information files.]

7. We note that you have included the phrase “data not shown” in your manuscript. Unfortunately, this does not meet our data sharing requirements. PLOS does not permit references to inaccessible data. We require that authors provide all relevant data within the paper, Supporting Information files, or in an acceptable, public repository. Please add a citation to support this phrase or upload the data that corresponds with these findings to a stable repository (such as Figshare or Dryad) and provide and URLs, DOIs, or accession numbers that may be used to access these data. Or, if the data are not a core part of the research being presented in your study, we ask that you remove the phrase that refers to these data.

Reviewers' comments:

Reviewer's Responses to Questions

**Comments to the Author**

1. Is the manuscript technically sound, and do the data support the conclusions?

Reviewer #1: Yes

Reviewer #2: Yes

2. Has the statistical analysis been performed appropriately and rigorously? 

Reviewer #1: Yes

Reviewer #2: Yes

3. Have the authors made all data underlying the findings in their manuscript fully available?

Reviewer #1: Yes

Reviewer #2: No

4. Is the manuscript presented in an intelligible fashion and written in standard English?

Reviewer #1: Yes

Reviewer #2: Yes

5. Review Comments to the Author

Reviewer #1: This article reports on the engineering and characterization of new yeast strains deleted for amino acid permeases to study transport properties and selectivity of these transporters. The engineering of such strains represents a major technical challenge, and the authors should be acknowledged for their achievement of this technical “tour de force”. The conclusions are supported by the data presented and can be of interest for scientists working on transport selectivity of amino acid transporters.

However, I believe there are major concerns that need to be address prior to publication at Plos One.

Major comments:

1. There should be a separate discussion section, not blended in with the results section. Please reformat the manuscript.

2. Regarding the discrepancy between the drop test assays (figure 2) and the radioactive uptake results (figure 3), I strongly recommend the authors to run a proteomic mass spectrometry analysis in the different strains and specifically look at protein levels of the remaining amino acid permeases still expressed in the strains. This will allow the authors to know whether or not there are some compensatory mechanisms (upregulation of certain transporters?) that could explain the discrepancy.

3. I am not sure if this is a compression problem that occurred during the submission process but please make sure to improve the resolution of all the figures (main and supplementary) if accepted for publication. As of now, the figures are very hard to read.

Minor comments:

1. Some sentences have grammatical errors and are hard to understand (line 80-83) ; even some words missing (line 45). Please correct.

2. Line 187: Could the authors comment in the text on their unsuccessful attempts to engineer additional deletions in the strain? Do they think it is due to synthetic lethality or technical problems?

3. Line 197: Add glu to the list of amino acids in which 22Δ8AA strain cannot grow.

4. For Fig S2: For the drop tests, I would have preferred to have the cells growing on the same plate for each amino acid tested instead of being patched together from three different plates. Right now, it raises suspicions as to whether or not the different dilution of cells were grown in the same conditions.

Reviewer #2: The manuscript details additional transporter gene deletions in classical yeast transporter mutants used in studies of amino acid transport. The authors generate strains with 11, 12, and 13 amino acid transporter deletions starting from a strain with 10 transporters deleted. The authors sequenced the genome of the parent strain and clarified a past error regarding the mating type of the host strain. Overall, the manuscript is concise and relatively clear.

Some critical issues:

1) The rationale for combinations of transporters deleted in 11, 12, and 13 deletion strains is not clear. The authors seemingly deleted random combinations of BAP3, BAP2, TAT1, TAT2, and AGP3. Why did the authors not pursue a more systematic or comprehensive deletion set? Why was BAP3 the only 11-deletion strain created and why not make all 5 11-deletion strains, especially considering the BAP3 11-deletion strain had a potential growth defect relative to 22Δ10α? The authors should clarify this rationale.

2) The authors claim the new strains will be useful for future studies of amino acid transport, yet the new strains did not exhibit significantly altered amino acid transport properties compared to the parental strain. What is the advantage of using the new strains over the classical and now sequenced 22Δ10α strain?

3) The study notes inconsistencies between growth assays and radiolabeled amino acid uptake. However, this finding seems unresolved. The authors should provide more speculation for this finding or provide additional assays to clarify this discrepancy.

Minor issues:

4) Page 12, Lines 232-235 – This sentence is very confusing. Why would deleting more amino acid transporters improve the uptake of amino acids?

5) Page 14, Lines 266-268 – Is it actually 48 contigs across 16 scaffolds?

6) Table 2 is confusing overall. Please clearly specify if all of the gene deletions/inactivations are in agreement with the expected mutations. Consider adding a column for observed deletion length. Were any deletions/inactivations different from what was expected? Were any adjacent genes or control elements affected? Would be helpful to stick to either bp or kbp for all lengths.

6. PLOS authors have the option to publish the peer review history of their article (what does this mean? ). If published, this will include your full peer review and any attached files.

**Do you want your identity to be public for this peer review?** For information about this choice, including consent withdrawal, please see our Privacy Policy .

Reviewer #1: No

Reviewer #2: No

---

## [Author Response · Author response to Decision Letter 0]

28 Feb 2025

Reviewer #1:

Major comments:

1. There should be a separate discussion section, not blended in with the results section. Please reformat the manuscript.

Response: Thank you for this suggestion. We have thoroughly revised the manuscript, and it now has a separate discussion that addresses the most relevant results.

2. Regarding the discrepancy between the drop test assays (figure 2) and the radioactive uptake results (figure 3), I strongly recommend the authors to run a proteomic mass spectrometry analysis in the different strains and specifically look at protein levels of the remaining amino acid permeases still expressed in the strains. This will allow the authors to know whether or not there are some compensatory mechanisms (upregulation of certain transporters?) that could explain the discrepancy.

Response: We agree that more work is needed to understand this discrepancy, and we really would have liked to address it. Unfortunately, due to limited resources and time, in addition to the fact that this falls out of the scope of this manuscript, we could not perform additional experiments (proteomics or at least transcriptomics) to explain this problem.

3. I am not sure if this is a compression problem that occurred during the submission process but please make sure to improve the resolution of all the figures (main and supplementary) if accepted for publication. As of now, the figures are very hard to read.

Response: We checked the uploaded figure files and compared it to the agglomerated pdf that was likely sent to the reviewers after processing by the PlosONE website, and we found that the original file has enough resolution and details to be legible with high magnification, while the one processed by the website has not (See Figure 1 below). We are not sure how to address this issue which is out of our control. In addition, we increased the resolution of S1 Fig, so that it is more legible. Thank you for pointing that out.

Minor comments:

1. Some sentences have grammatical errors and are hard to understand (line 80-83) ; even some words missing (line 45). Please correct.

Response: We apologize for this oversight. We carefully checked the manuscript, and those, in addition to others, should now be corrected.

2. Line 187: Could the authors comment in the text on their unsuccessful attempts to engineer additional deletions in the strain? Do they think it is due to synthetic lethality or technical problems?

Response: We added a few sentences to explain the reason we did not pursue this further: Lines 335-344 in the discussion. “We initially followed up with deleting additional genes and intended to delete TAT2 from 22∆13 as an avenue to decrease the background growth further, but with no success. Because the kanMX cassette had not been removed from the 22∆13 genome, we used the natNT2 cassette which provides resistance to the antibiotic cloNAT [56], thereby skipping one step in the process of creating higher order mutants. TAT2 deletion leads to viable strains, proven by the viability of 22∆12’. It is possible that our conditions were not optimal for selecting the cells which successfully inserted the cassette into TAT2, or that the double mutant TAT1 TAT2 in the 22∆11background is not viable. We did not pursue this further because the higher order deletion mutants 22∆11, 22∆12, 22∆12’ and 22∆13 did not show noticeable growth differences compared to 22∆10α on selective amino acid media (Fig 2).”

4. Line 197: Add glu to the list of amino acids in which 22Δ8AA strain cannot grow.

Response: Corrected. Thank you for noticing this.

3. For Fig S2: For the drop tests, I would have preferred to have the cells growing on the same plate for each amino acid tested instead of being patched together from three different plates. Right now, it raises suspicions as to whether or not the different dilution of cells were grown in the same conditions.

Response: We agree that the layout of this figure looks suspicious. We decided to design it that way because, for some amino acids, the rows of yeast dilutions were dropped a little far apart, which would have made a too large figure. We added the text “For each amino acid / concentration combination, the dilutions of cells were dropped on a single Petri dish, despite them being shown as strips” in the legend. See also Fig 2 below for a comparison of a panel of the figure and the original Petri dish scan. We would be happy to add the original scans in the supporting information if desired by the reviewer.

Reviewer #2:

Some critical issues:

1) The rationale for combinations of transporters deleted in 11, 12, and 13 deletion strains is not clear. The authors seemingly deleted random combinations of BAP3, BAP2, TAT1, TAT2, and AGP3. Why did the authors not pursue a more systematic or comprehensive deletion set? Why was BAP3 the only 11-deletion strain created and why not make all 5 11-deletion strains, especially considering the BAP3 11-deletion strain had a potential growth defect relative to 22Δ10α? The authors should clarify this rationale.

Response: We understand that, as described, our work seems to lack logic. Our initial goal was to delete 5 more amino acid permeases from 22∆10α, namely Bap2, Bap3, Tat1, Tat2 and Agp3, the remaining major amino acid permeases (using the data published in Regenberg, 1999). We started to delete in parallel BAP3 and BAP2-TAT1, but only the former was successful at the first attempt, leading to 22∆11. We then used this strain to delete in parallel the other genes, leading to 22∆12, 22∆12’, 22∆13. As mentioned in the text (lines 345-347), the slow growth and the fact that further deletions did not lower much the background of 22∆10α, we did not attempt more deletions. It was much later that we found that some 22∆11 siblings did not display this slow growth phenotype, and, because of time and funding constraints, we could not repeat the deletion process from one of those siblings.

2) The authors claim the new strains will be useful for future studies of amino acid transport, yet the new strains did not exhibit significantly altered amino acid transport properties compared to the parental strain. What is the advantage of using the new strains over the classical and now sequenced 22Δ10α strain?

Response: We are now discussing this point in the new discussion (lines 361-367). Hopefully, it now addresses the reviewer’s comment.

3) The study notes inconsistencies between growth assays and radiolabeled amino acid uptake. However, this finding seems unresolved. The authors should provide more speculation for this finding or provide additional assays to clarify this discrepancy.

Response: We have included ample speculation in the discussion (lines 395-431). Hopefully, this addresses the reviewer’s comment.

Minor issues:

4) Page 12, Lines 232-235 – This sentence is very confusing. Why would deleting more amino acid transporters improve the uptake of amino acids?

Response: this sentence has been deleted, as part of the separation of the discussion from the results.

5) Page 14, Lines 266-268 – Is it actually 48 contigs across 16 scaffolds?

Response: Our assembly led to 25 contigs, which were assembled in 48 scaffold, as now explained in the text: “Using 8.12 Gbp of long-read PacBio sequencing data, the 22∆10α genome was assembled de novo using Canu to an estimated depth of 600x [31], leading to 25 contigs. Aligning the contigs to the S288C reference genome showed coverage over all chromosomes further supporting the assembly (Fig 4a). Furthermore, after reference-guided separation of erroneously collapsed contigs, scaffolding and polishing, no large duplications or translocations were identified (Fig 4b), leading to a 12.87 Mbp assembly in 48 scaffolds. Noticeably, 95% of the assembly was contained within the largest 16 scaffolds (12.2 Mbp, corresponding to the entire yeast genome), with the remaining 5% corresponding to totally ordered contigs not assembled into scaffolds, and aligning with miscellaneous regions of the genome (S3 Fig).”

6) Table 2 is confusing overall. Please clearly specify if all of the gene deletions/inactivations are in agreement with the expected mutations. Consider adding a column for observed deletion length. Were any deletions/inactivations different from what was expected? Were any adjacent genes or control elements affected? Would be helpful to stick to either bp or kbp for all lengths.

Response: Thank you for this suggestion. We have revised Table 2, providing more details about the expected (when known) and observed mutations. We also modified the corresponding text in the results, clarifying that the nature of most of the mutations were not known – only that of the mutations in GNP1 and AGP1, created by our lab, were known (L274-276).

---

## [Decision Letter · Decision Letter 1]

17 Mar 2025

Characterization and whole genome sequencing of Saccharomyces cerevisiae strains lacking several amino acid transporters: tools for studying amino acid transport

PONE-D-24-55165R1

Dear Dr. Pilot,

We’re pleased to inform you that your manuscript has been judged scientifically suitable for publication and will be formally accepted for publication once it meets all outstanding technical requirements.

Kind regards,

Patrick Lajoie, PhD

Academic Editor

PLOS ONE

Reviewers' comments:

Reviewer's Responses to Questions

**Comments to the Author**

1. If the authors have adequately addressed your comments raised in a previous round of review and you feel that this manuscript is now acceptable for publication, you may indicate that here to bypass the “Comments to the Author” section, enter your conflict of interest statement in the “Confidential to Editor” section, and submit your "Accept" recommendation.

Reviewer #1: All comments have been addressed

Reviewer #2: All comments have been addressed

2. Is the manuscript technically sound, and do the data support the conclusions?

Reviewer #1: Yes

Reviewer #2: Yes

3. Has the statistical analysis been performed appropriately and rigorously? 

Reviewer #1: Yes

Reviewer #2: Yes

4. Have the authors made all data underlying the findings in their manuscript fully available?

Reviewer #1: Yes

Reviewer #2: Yes

5. Is the manuscript presented in an intelligible fashion and written in standard English?

Reviewer #1: Yes

Reviewer #2: Yes

6. Review Comments to the Author

Reviewer #1: The manuscript has improved quite a lot compared to the original submission.

The authors have answered all the questions/comments.

The discussion is much more developed and brings useful insights into potential limitations of the study.

Reviewer #2: The authors have adequately addressed prior reviewers comments and improved the manuscript for publication

7. PLOS authors have the option to publish the peer review history of their article (what does this mean? ). If published, this will include your full peer review and any attached files.

**Do you want your identity to be public for this peer review?** For information about this choice, including consent withdrawal, please see our Privacy Policy .

Reviewer #1: No

Reviewer #2: No

---

## [Editor Report · Acceptance letter]

PONE-D-24-55165R1

PLOS ONE

Dear Dr. Pilot,

I'm pleased to inform you that your manuscript has been deemed suitable for publication in PLOS ONE. Congratulations! Your manuscript is now being handed over to our production team.

Kind regards,

on behalf of

Dr. Patrick Lajoie

Academic Editor

PLOS ONE